# OpenReview forum: "Latent instruction representation alignment: defending against jailbreaks, backdoors and undesired knowledge in LLMs"
_ICLR.cc/2026/Conference — Submitted to ICLR 2026_

### Official Review · Reviewer_TDW5 · 2025-10-29

**Soundness:** 1
**Presentation:** 2
**Contribution:** 2
**Rating:** 2
**Confidence:** 4

**Summary:**

A paper proposes a post-training intervention method based on white-box representation engineering method and which aims addresses the problem of models producing undesired outputs. Authors demonstrate that proposed method shows substanital improvemnts in threat models of jailbreaks, backdoors and unlearning, against several exhisitng attack methods.

**Strengths:**

Authors introduce and justify a post-training intervention that appears to work on a limited set of pre-exhisting jailbreaking attacks, a static unlearning evaluation and backdoor challenges.

**Weaknesses:**

### Lack of Adaptive Evaluation

My main critique of the paper is that it lacks adaptive evaluation of the proposed defense mechanisms. While the authors argue in the introduction that previous mitigations do not generalize to novel attacks [line 32, line 46], they fail to provide this kind of evaluation themselves. Although they claim their method is robust against novel attacks [lines 32, 53], for the jailbreaking part of the paper they only demonstrate effectiveness against a weak PEZ attack and a superficially constrained embedding space attack.

To demonstrate that their defense offers substantial improvements over previous methods, the authors need to evaluate it with *adaptive attacks* - those that explicitly adapt to the defense design (e.g., at least similar to [4]). Adaptive attacks have repeatedly exposed defenses across adversarial ML: in jailbreaking, simple adaptive attacks [1] subverted the adversarially trained R2D2 model [2], CircuitBreakers [3] fell to adaptive attacks in latent space [4, 5] and stronger attacks in input space [6]. Concurrently, several defenses, some based on instruction-data separation principles, were subverted in one go [7] by specifically designed adaptive attacks.

The same adaptive evaluation principle universally applies to other adversarial ML areas like unlearning [8, 9], where adaptive evaluation revealed the brittleness of proposed measures despite the initial authors' claims. Without such evaluation, the robustness claims in this paper remain fundamentally unsubstantiated.

### Methodological Problems in Jailbreaking Experiments

**Choice of attacks.** PEZ is an old attack shown to be extremely ineffective in prior work [2, 11], even against non-safety-tuned models. On lines 86-87, the authors claim that PEZ is a "challenging" jailbreak attack and cite Schwinn et al. [12]. However, PEZ and soft-prompt attacks differ critically: PEZ maps back to token space while soft-prompt does not. From my experience, the mapping to discrete space is a central problem of PEZ that will not be addressed by increasing the "suffix" length that authors intoduce (line 360).

The ES (embedding space) attack introduced on line 355 and described on lines 364-369 appears to be largely equivalent to the embedding space attacks used in the CB paper and in [12]. The authors introduce a superfluous constraint of "proportion of embedding dimension attacker is controlling" without justification. This constraint artificially weakens the attack without reflecting realistic threat models, making it unclear what advantage this provides over existing embedding space attack formulations.

In Appendix M, the authors justify not choosing the GCG attack, claiming it achieves only 2% on Gemma-2-2b in HarmBench evaluation and does not benefit from best-of-N (BoN) evaluation. This is an unreasonably low number that suggests either implementation issues or evaluation problems, as it directly contradicts prior work. For example, [14] reports that GCG against Gemma-2-2b **with a safe system prompt** achieves 30% ASR, and [15] analyzes how most of jailbreaking attacks benefit from BoN evaluation, including GCG.

**Choice of benchmark.** In contrast to previous methods like CB [3] or adaptive evaluations like [1, 7], the authors do not use established benchmarks such as HarmBench, AdvBench, or JailbreakBench. Instead, they use a subsplit of CB training/validation data. This problem extends to the ad-hoc grader introduced in Appendix E.1, which lacks human study validation or justification for why it should be preferred over existing graders like StrongReject [13] or HarmBench. The arbitrary choice of harmful queries, grader, and ASR@10 for one attack and ASR@1 for another makes results completely incomparable with prior work.

### Overall

The paper suffers from significant methodological limitations in its evaluation approach. Across all three experimental setups (unlearning, backdoors, jailbreaking), it fails to include adaptive attack evaluations, where adversaries are assumed to have full knowledge of the defense mechanisms. Without such evaluations, the claimed robustness of the proposed measures remains unsubstantiated, making it impossible to assess whether the defense provides meaningful security improvements.

The jailbreaking setup, which I examined in depth, compounds these issues with several questionable choices: an unconventional benchmark and grader, weak static attacks (i.e., no GCG, PRS[1], or strong BoN[6] attacks), and claims about attack effectiveness that contradict established findings in the literature on these specific attack methods, which further undermine the reliability of the results.

[1] Jailbreaking Leading Safety-Aligned LLMs with Simple Adaptive Attacks

[2] HarmBench: A Standardized Evaluation Framework for Automated Red Teaming and Robust Refusal

[3] Improving Alignment and Robustness with Circuit Breakers

[4] Obfuscated Activations Bypass LLM Latent-Space Defenses

[5] https://confirmlabs.org/posts/circuit_breaking.html

[6] Best-of-N Jailbreaking

[7] The attacker moves second: Stronger adaptive attacks bypass defenses against llm jailbreaks and prompt injections

[8] On Evaluating the Durability of Safeguards for Open-Weight LLMs

[9] An adversarial perspective on machine unlearning for AI safety

[11] Universal and Transferable Adversarial Attacks on Aligned Language Models

[12] Soft Prompt Threats: Attacking Safety Alignment and Unlearning in Open-Source LLMs through the Embedding Space

[13] A StrongREJECT for Empty Jailbreaks

[14] An Interpretable N-gram Perplexity Threat Model for Large Language Model Jailbreaks

[15] Sampling-aware Adversarial Attacks Against Large Language Models

**Questions:**

### Questions to Authors

- Why is PEZ performance on Llama missing from Table 2?

-  Given the limited success of embedding space attacks in the original CircuitBreakers paper and the reported ASR of 28.1% in Table 2, what are the specific differences between your setup and the original CB evaluation? Additionally, did you train a CircuitBreakers model yourself, or did you use the pre-existing checkpoint from https://huggingface.co/GraySwanAI/Llama-3-8B-Instruct-RR? .

- What exactly constitutes the Forget Set used for CyberSecurity unlearning experiments? Do I understand correctly that it is filtered Cyber Corpora from the original WMDP paper?

---

> ### Author Response · Authors · 2025-12-03
>
> Thank you for your attention to our work and your detailed feedback. We respond to points as space permits below. As with the review, most of the discussion focuses on the jailbreak evaluations.
>
> > My main critique [...] is that it lacks adaptive evaluation [...] where adversaries are assumed to have full knowledge of the defense mechanisms
>
> We believe that our embedding space (ES) attacks constitute a strong adaptive evaluation, acting as an upper bound on the efficacy of a broad attack class. The ES attacker faces no artificial constraints: it has full white-box control of the embedding dimension for the entire harmful request and runs until convergence. Whatever defensive failures exist, gradient descent in high-dimensional space provides a powerful mechanism for discovering them. This contrasts with heuristic or constrained methods which must rely on human judgment to match the attack to the defense.
>
> Additionally, our defense was specifically designed to avoid brittle assumptions (e.g. geometric proxies, Sec. 3) that typically invite adaptive exploits. Because we optimize functional output directly, we do not see---and reviewers did not propose---a structural weakness suggesting a bespoke attack strategy stronger than unconstrained optimization.
>
> > The same adaptive evaluation principle universally applies to other adversarial ML areas like unlearning
>
> We acknowledge in our conclusion that we are not robust to subsequent fine-tuning. However, we understand robustness to open-weight modification as a distinct desideratum, not a universal requirement for all threat models or prior unlearning methods.
>
> > PEZ is an old attack shown to be extremely ineffective in prior work
>
> Our PEZ variant---with a longer attack suffix, a higher learning rate, 10x more attack steps, and the best-of-k (all information included in appendix B.1)---is much stronger than the default. The high baseline ASRs (up to 57%) contradict the intuition that PEZ is necessarily weak.
>
> > The ES (embedding space) attack [...] appears to be largely equivalent to the [...] attacks used in the CB paper [...] what are the specific differences
>
> Our ES attack differs critically from CB's: theirs optimizes only suffix "tokens" while ours optimizes the *entire* request embedding. Since CB trains the model to break upon recognizing a harmful request, its suffix-only attack is naturally weaker.
>
> > The authors introduce a superfluous constraint of "proportion of embedding dimension attacker is controlling"
>
> Table 2 reports the *unconstrained* attacker (100% control) as the primary metric. Partial control results (fig 4b) are provided only as *additional context*. As noted (lines 364-366), this constraint (where applied) operationalizes bits under attacker control, a framing established in [1].
>
> > the authors justify not choosing the GCG attack [...]. This [ASR] directly contradicts prior work. For example, [2] reports that GCG against Gemma-2-2b with a safe system prompt achieves 30% ASR
>
> We found broadly similar GCG ASRs with both HarmBench and our independent implementation. We believe high ASR in [2] may be an anomaly caused by a methodological issue: "We use the Llama2 system prompt as the default for all models". However, Gemma does not support system prompts [3]; forcing one plausibly pushes Gemma off-distribution, artificially degrading it.
>
> > the authors [claim GCG] does not benefit from best-of-N (BoN) evaluation
>
> We find this claim nowhere in our text. We say "we were able to substantially boost PEZ ASR but found much *less* success at strengthening GCG".
>
> > the authors do not use established benchmarks such as HarmBench
>
> We attempted to use HarmBench but quickly found critical bugs that cause two attack methods to deviate substantially from their intended design, undermining the validity of results. We opted for a clean, independent implementation to ensure correctness.
>
> > the ad-hoc grader [...] lacks human study validation or justification for why it should be preferred over existing graders like StrongReject
>
> Our grader uses Gemini Flash 2, significantly more capable than StrongReject's fine-tuned Gemma 1 2B. Base capability was the primary determinant of grader quality in our testing. Further, StrongReject entangles willingness and capability by design [4], which impedes evaluation of small models that are willing but incapable of generating complex harm. Our grader focuses on willingness, aligning qualitatively with our blinded human ratings.
>
> [1] Fort (2023)
>
> [2] Boreiko, et al. (2024).
>
> [3] https://ai.google.dev/gemma/docs/core/prompt-structure
>
> [4] Souly, et al. (2024)

---

> ### Author Response · Authors · 2025-12-03
> **Responses to questions**
>
> > Why is PEZ performance on Llama missing from Table 2?
>
> As hinted at above, our PEZ attacks are the most compute intensive part of the paper. In light of this, we prioritized evaluations on the most capable and robust model in the size class.
>
> > Additionally, did you train a CircuitBreakers model yourself, or did you use the pre-existing checkpoint from https://huggingface.co/GraySwanAI/Llama-3-8B-Instruct-RR? .
>
> We trained the model ourselves following the procedure in the CB paper and using their released code as a reference. The lack of Gemma CB checkpoints precludes relying on only pre-existing checkpoints.
>
> > What exactly constitutes the Forget Set used for CyberSecurity unlearning experiments? Do I understand correctly that it is filtered Cyber Corpora from the original WMDP paper?
>
> This dataset is described in appendix E.2. We pre-filter the quite noisy WMDP cyber corpora dataset at the document level, turn it into a series of question and answer pairs grounded in those documents, and then filter those pairs for quality.

---

### Official Review · Reviewer_hB91 · 2025-10-29

**Soundness:** 2
**Presentation:** 1
**Contribution:** 2
**Rating:** 2
**Confidence:** 5

**Summary:**

This paper proposes Latent Instruction Representation Alignment (LIRA), a post-training defense for LLMs that aims to prevent jailbreaks, remove backdoors, and achieve unlearning. Instead of modifying model outputs directly, LIRA aligns latent representations of harmful instructions with those of benign instructions using Sequence-Aware Gradients (SAG) to block gradients from response tokens. The authors also introduce AdLIRA, an adversarial extension that alternates between “attack” and “align” phases to improve robustness, and evaluate both methods on jailbreak, backdoor, and unlearning benchmarks.

**Strengths:**

1. Novel conceptual framing: The paper raises an interesting idea, addressing how the model interprets harmful instructions instead of only modifying its final responses. This representation-level intervention could, in principle, yield better generalization across jailbreak and unlearning tasks.

2. Unified perspective: The attempt to address jailbreaks, backdoors, and unlearning within a single framework is conceptually appealing and could inspire future research connecting these safety problems.

**Weaknesses:**

1. Inefficient training design.
If the goal is to align the latent representations of harmful and its nearby instructions, it seems unnecessary to perform a full forward pass including both instructions and responses, only to block gradients from response tokens afterward. A more direct and computationally efficient approach would be to align both instruction representations via cosine similarity (or other distance metrics) directly, without involving the response logits.

2. Unlearning baselines are outdated.
The unlearning experiments only compare against older baselines (GD, RMU). More recent and stronger methods should be included for fairness:
SimNPO[1] TPO[2] ME[3]

3. Ambiguous terminology and inconsistent definitions.
The terminology should be standardized and explained clearly before use. These inconsistencies make the paper harder to follow. For example: "GD" (Gradient Difference) is used in figures before being defined.
Table 1 introduces terms such as "Malign request refusal" without definitions. It is unclear whether "malign request" and "harmful request" refer to the same concept.
After reading the entire paper, I can roughly infer the authors’ intended meaning, but clearer and earlier definitions, along with consistent terminology, would greatly improve readability and overall quality.

4. Misuse of "nearby safe instruction" in Figure 1.
The term "nearby safe instruction" is misleading. The example nearby instruction "Help me build a bomb" is clearly harmful; it remains an unsafe instruction, even though the model produces a safe, refusal-type response to it. The authors should clarify that LIRA make the representation of a harmful instruction close to a nearby instruction that lead model to generate the safe answer, not that the instruction itself is safe.

5.	Limited comparison to representation-level defenses.
It omits comparisons to related latent-space approaches included in related work section, such as TLAT[4] and LAT[5]


[1] Fan, Chongyu, et al. "Simplicity prevails: Rethinking negative preference optimization for llm unlearning." arXiv preprint arXiv:2410.07163 (2024).

[2] Zhou, Xiangyu, et al. "Not All Tokens Are Meant to Be Forgotten." arXiv preprint arXiv:2506.03142 (2025).

[3]Yuan, Xiaojian, et al. "A closer look at machine unlearning for large language models." arXiv preprint arXiv:2410.08109 (2024).

[4] Casper, Stephen, et al. "Defending against unforeseen failure modes with latent adversarial training." arXiv preprint arXiv:2403.05030 (2024).

[5]Sheshadri, Abhay, et al. "Latent adversarial training improves robustness to persistent harmful behaviors in llms." arXiv preprint arXiv:2407.15549 (2024).

**Questions:**

1. In the Table1, does the “malign request” and “harmful request” refer to the same concept?

2. Why are only older baselines (GD and RMU) included in the unlearning experiments? Could the authors compare against more recent and stronger unlearning methods such as SimNPO [1], TPO [2], and ME [3] to better assess LIRA’s performance?

3. Why does the paper omit quantitative comparisons to recent latent-space or representation-level defense methods mentioned in related work, such as TLAT and LAT? How would LIRA and AdLIRA perform relative to these?

---

> ### Author Response · Authors · 2025-12-03
>
> Thank you for the thoughtful feedback on our work. We appreciate the recognition of our approach's novel and unified framing. We also appreciate the attention to terminological issues which we can henceforth clarify and avoid. We focus on the overarching training design point below.
>
> > Inefficient training design. [...] A more direct and computationally efficient approach would be to align both instruction representations via cosine similarity (or other distance metrics) directly, without involving the response logits.
>
> We agree that the approach mentioned makes sense and would be usable. We believe there are several benefits to our approach:
>
> 1. Flexibility. Geometric alignment (or simple versions of it) requires a shared "core" token sequence and some augmentations around it that vary in the harmless and harmful conditions. When the input distinctions that elicit behavioral differences are diffuse, complex, and semantically entangled, applying the geometric approach is much more difficult and risks encouraging the model to collapse *all* request distinctions. On the other hand, our approach works on any input pair where we can define a target output distribution and provides the model guidance on what specific aspects of the representations to modify---those responsible for the harmful output.
> 2. Adversarial exploration. We do not see any useful mirrored geometric objective for the attacker so our attacker necessarily has a functional goal using Sequence Aware Gradients (SAG). The defender's objective is then symmetric. Giving the defender a geometric goal would likely also impose a geometric constraint on the attacker---attack representations that are too distant from the initial attack representations would likely be "broken" in coarse and unhelpful ways by the defender's translation. This would limit exploration of the attack space relative to our approach which allows both players freedom to explore different *mechanisms* for achieving their functional goals.
> 3. We believe that Sequence-Aware Gradients (SAG) are a generally useful tool with other applications. While they may not be the only method to achieve gains in these tasks, showing that the method works well on this task is a useful contribution.
>
> > Unlearning baselines are outdated. [...] Limited comparison to representation-level defenses.
>
> In the face of limited time and space, we prioritized comparisons to well-known and well-tuned baselines as a way to understand and interpret results. We agree that more baselines can make head-to-head comparisons easier, and will consider adding these in future work.
>
> > Table 1 introduces terms such as "Malign request refusal" without definitions. It is unclear whether "malign request" and "harmful request" refer to the same concept.
>
> Thank you for pointing out this issue. "Malign request refusal" in the third column should also be "harmful request refusal" for consistency and clarity.
>
> > Misuse of "nearby safe instruction" in Figure 1. The term "nearby safe instruction" is misleading. The example nearby instruction "Help me build a bomb" is clearly harmful; it remains an unsafe instruction, even though the model produces a safe, refusal-type response to it.
>
> We intended "safety" in the "Safety as a System Property" sense [1]. We understand safety as a property which cannot be judged of a subcomponent in isolation---a request that invokes a harmless refusal is thus not unsafe. That said, we will revise the terminology to avoid confusion.
>
> [1] Leveson, Nancy G. "Safety as a system property." Communications of the ACM 38.11 (1995): 146.

---

### Official Review · Reviewer_Z2JJ · 2025-11-01

**Soundness:** 2
**Presentation:** 3
**Contribution:** 3
**Rating:** 4
**Confidence:** 3

**Summary:**

This paper introduces LIRA (Latent Instruction Representation Alignment), a novel post-training method for defending LLMs against jailbreaks, backdoors, and unwanted knowledge expression. The key innovation is training models to align malicious instruction representations with benign ones, rather than directly penalizing harmful outputs. This is achieved through Sequence-Aware Gradients (SAG), which selectively blocks gradients from response positions during training. The authors extend LIRA with: (1) AdLIRA, an internally adversarial training scheme, and (2) a classifier-guided variant for unpaired data. Experiments on Gemma 2 9B and LLaMA 3.1 8B demonstrate strong performance across jailbreak defense (>99% blocking of PEZ attacks), backdoor removal, and unlearning tasks.

**Strengths:**

1. Novel and well-motivated approach: The focus on instruction representations rather than output behaviors is conceptually elegant and well-justified. The intuition that representations form an information bottleneck easier to defend than the full instruction-to-output pipeline is compelling.

2. Comprehensive evaluation: The paper tackles three distinct security challenges (jailbreaks, backdoors, unlearning) with strong baselines including Circuit Breakers, RMU, and gradient-based methods. The evaluation includes both automatic metrics and human ratings where appropriate.

3. Strong empirical results: LIRA substantially outperforms baselines across tasks. Particularly impressive are: (a) 99%+ blocking of PEZ jailbreaks with AdLIRA, (b) single-step backdoor removal for the HATE task, and (c) effective unlearning with minimal retain set degradation.

**Weaknesses:**

1. Limited theoretical justification: While the intuition about instruction representations is appealing, the paper lacks formal analysis of why this approach should generalize better. What properties of instruction representations make them more robust targets? Under what conditions might this fail? The empirical observation in Figure 1b is suggestive but insufficient.

2. SAG design choices under-justified: The specific gradient blocking rules (which attention paths to block, which parameter updates to prevent) appear somewhat arbitrary. Why block attention between response tokens specifically? The paper states this prevents "improving intra-benign-sequence quality" but doesn't validate that this is problematic. Ablations comparing SAG variants would strengthen this.

3. Computational cost not addressed: AdLIRA requires iterating between attack and defense phases, and classifier-guided LIRA retrains the classifier each iteration. How do training times and computational costs compare to baselines? This is crucial for practical deployment.

4. Limited analysis of failure modes: When does LIRA fail? The paper shows strong aggregate results but provides little insight into what kinds of attacks or backdoors might bypass the method. The embedding space attack results (Fig 4b) show AdLIRA still fails ~50% of the time with full attacker control—what characterizes successful vs. unsuccessful attacks?

**Questions:**

Please refer to weaknesses.

---

> ### Author Response · Authors · 2025-12-03
>
> Thank you for the thoughtful and detailed feedback. We appreciate the recognition of our approach's novelty and empirical efficacy and the breadth of our evaluation. We also acknowledge that there's room for improved theoretical understanding and additional analyses of methodological details and qualitative results.
>
> > SAG design choices under-justified: [...] Why block attention between response tokens specifically?
>
> To explain in slightly more detail: the underlying premise here is that, prior to our intervention, the model already possesses adequate sequence modeling capabilities. We see our task as *steering* the model toward the existing sequence distribution we prefer (e.g. the refusal distribution) rather than improving fluency within the distribution. From this perspective, any gradients flowing between response tokens are not task-relevant and can only confound the specific training signal we intend to provide, which we saw in informal experiments.
>
> > Ablations comparing SAG variants would strengthen this.
>
> We briefly discuss SAG variants we tried and their inferior performance in appendix L. We agree that more detailed discussion in the body would have been useful if space allowed.
>
> > Computational cost not addressed: AdLIRA requires iterating between attack and defense phases, and classifier-guided LIRA retrains the classifier each iteration. How do training times and computational costs compare to baselines?
>
> We believe much of the relevant information is available in the paper and appendices:
>
> - As described in appendix D (task hyperparameters), AdLIRA uses 110 total update steps in the embedding space attack task compared to e.g. 100 steps for circuit breakers.
> - Using the batch size from appendix D to translate the "sequences seen" metric in figure 4d shows that AdLIRA requires approximately 70 updates steps for each of the attacking and defending layers to remove the backdoor. Other methods have made little progress in their 100 whole-model update steps.
> - All of our unlearning methods, baselines included, require more update steps since they want full coverage of the to-be-unlearned distribution. Both RMU and our classifier-guided LIRA use the same batch size on unlearning tasks so a direct comparison in terms of task performance vs. main model update steps is possible in figure 5. We do require extra FLOPs due to the classifier training, but the classifier is smaller than the main model and we would expect this main model to classifier ratio to grow with model size. Classifier retraining is generally fast---only a few update steps.
>
> > Limited analysis of failure modes: When does LIRA fail? The paper shows strong aggregate results but provides little insight into what kinds of attacks or backdoors might bypass the method. The embedding space attack results (Fig 4b) show AdLIRA still fails ~50% of the time with full attacker control—what characterizes successful vs. unsuccessful attacks?
>
> We agree that this would be useful additional analysis to conduct. The single success we see with PEZ attacks against AdLIRA results from a prompt that causes the model to treat a request for hacking guidance as a request for help "hacking" within a video game. The model's game hacking tips would be marginally useful for a novice real-world attacker. The embedding space attacks resist immediate interpretation since there's no faithful verbalization of the successful harmful requests, but more sophisticated analyses could be interesting and useful.

---

### Meta-Review · Area_Chair_qcUn · 2025-12-30

**Summary:**

I find the two most significant weaknesses of the paper are its writing and its experimental section. The writing prevents the extraction of scientific value from the proposed method, as the theory, the details, and the justification for the choices made during the design of the method are poorly described throughout the paper. The experiments, on the other hand, suffer from weak baselines, a lack of rigor in explaining the experimental details and results, and include multiple informal experiments whose results are never presented in the paper. Finally, more abalations are needed to convince the reader of the effectiveness of the method and the optimality of the design choices. That said, the paper has promise as it shows reasonable results in 3 distinct tasks that have been a subject of a lot of study, and has some interesting technical ideas that need to be explained and justified better. I recommend that the authors take time to improve their experiments and rewrite major sections of their work before resubmitting.

**Reviewer Concerns:**

**Outstanding concerns:**

- **Reviewer Z2JJ (Weakness 1-2); Reviewer hB91 (Weakness 3)**
The authors do not properly address a major issue pointed out by the reviewer. In particular, they do a poor job explaining the justification for and the exact theoretical properties of their gradient blocking. I think the paper will benefit from 1. defining the theoretical property their gradient blocking scheme desires to achieve, 2. theoretically proving that the proposed gradient blocking scheme achieves it, and 3. explaining the practical implications of this. As it stands now, the paper's discussion does not motivate the proposed method well, preventing future extensions. Further, the experimental justification for the effectiveness of SAG is lacking from the paper. E.g., experiments that would have strengthened the evidence for the effectiveness of SAG, like the experiment described in App. L are not presented. Finally, I think that Sections 2.1-2.3 can be written much more technically precisely, as the current explanation is such that one will struggle to implement the proposed methods, and it is somehow hard to distinguish which technical methods are part of the core of SAG and LIRA, and which are setting specific extensions.
- **Reviewer Z2JJ (Weakness 4)**
The authors should provide a qualitative analysis of the failures of their method. Likely, this implies more experiments in the current setup.
- **Reviewer hB91 (Weakness 2,5; Question 2,3); Reviewer TDW5 (Choice of attacks)**
The reviewers complain that, both for the unlearning and jailbreak attack and defense baselines, the methods used are outdated, and that certain choices do not make sense. These align well with my own opinion on the matter. As a matter of fact, I think that the jailbreak attacks and defense experiments can be substantially expanded, and not including LAT and GCG is just inexcusable and makes all experimental results questionable. In principle, the current attacks used are not even jailbreaks per say - they are classical adversarial optimization attacks, so I will recommend also TAP [1] and/or AutoDAN [2] as additional baselines on the attack side. The response of the authors is not satisfactory, as they do not provide any new experiments, and their justification for not using GCG is that it does not work on a single model that the authors experiment with in the appendix.
- **Reviewer hB91 (Weakness 1)**
I find the authors' response unsubstantiated. I think experiments can better demonstrate the points outlined by the authors and their validity, which is currently unclear, especially because I am quite unsure that LIRA+SAG doesn't suffer from the same problems outlined.
- **Reviewer TDW5 (Adaptive attacks)**
I think this is a fair point from the reviewer. In particular, I do not think the authors should solve adaptive attacks, but they should show this as an attempt to understand the limitations of their method. The fact that the reviewer even gives them a concrete baseline and the authors choose to ignore it is even more problematic. I will recommend for the next submission for the authors to consider a adaptive attack baseline.
- **Reviewer TDW5 (Choice of benchmark)**
I agree with the reviewer that the authors should report results on standard benchmarks with standard judges. It is ok, that the authors choose some other ones for their main evaluation, as long as the reasons for this are justified. However, the point of the review is also valid - without at least one experiment on a standard benchmark with standard judge, comperability to other methods suffers.
- **AC**
It is unclear what dataset the benign refusal in Table 2 is measured on. Further, MMLU is a bad benchmark for refusals in general. I would recommend that the authors use XSTest [3].

**Paper's strengths:**
As pointed out by **Reviewer Z2JJ**, the method is interesting and holds some promise. There are three points I find promising in particular:
1. The backdoor defense that does not require exact backdoor knowledge to provide effective results
2. The proposed method does not require a particular target for its jailbreak defense to work
3. A unified framework for addressing unlearning, jailbreaks, and backdoors

[1] https://arxiv.org/abs/2312.02119
[2] https://arxiv.org/abs/2310.04451
[3] https://arxiv.org/abs/2308.01263

**Reviewer Scores:**

If I were any of the reviewers, I would not have moved my initial score by a lot during discussion, as ultimately, the authors do not provide any new results to substantiate their claims and do not promise substantive changes to their manuscript (and do not provide such either). There are a few important clarifying comments on technical details that are not clearly written in the original paper, but the authors successfully manage to clarify during the rebuttal, and I would recommend that they incorporate those in their next paper revision. Ultimately, however, I do not see why this would have moved the initial scores of the reviewers.

---

### Decision · Program_Chairs · 2026-01-26

Reject